# Radiotherapy as a New Player in Immuno-Oncology

**DOI:** 10.3390/cancers10120515

**Published:** 2018-12-14

**Authors:** Shang-Jui Wang, Bruce Haffty

**Affiliations:** Department of Radiation Oncology, Rutgers Cancer Institute of New Jersey, 195 Little Albany St., New Brunswick, NJ 08901, USA; sjwang0416@gmail.com

**Keywords:** radiotherapy, radiation, immunotherapy, combination therapy, cancer, dose-fractionation, timing

## Abstract

Recent development in radiation biology has revealed potent immunogenic properties of radiotherapy in cancer treatments. However, antitumor immune effects of radiotherapy are limited by the concomitant induction of radiation-dependent immunosuppressive effects. In the growing era of immunotherapy, combining radiotherapy with immunomodulating agents has demonstrated enhancement of radiation-induced antitumor immune activation that correlated with improved treatment outcomes. Yet, how to optimally deliver combination therapy regarding dose-fractionation and timing of radiotherapy is largely unknown. Future prospective testing to fine-tune this promising combination of radiotherapy and immunotherapy is warranted.

## 1. Introduction

The notion that the sole driver of carcinogenesis lies upon genomic instability and oncogenic mutations has long been relinquished. It is well established that tumor immunology involving the three phases of immune response against cancer cells (the three “E”s—elimination, equilibrium, and escape) plays a major role in tumor formation. The immune system is the first line of defense against cells that have transformed and gone rogue via elimination. Cancer cells that have evaded the initial immune assault continue to evolve by Darwinian selection against pressure exerted by the immune system. Ultimately, the final phase of escape from immune surveillance allows the cancer cells to thrive unchecked in an immunosuppressive tumor microenvironment (TME). This has redefined our perception on combating malignancy, as the art of cancer therapy no longer just involves the surgical removal of tumor or destruction of cancer cells by means of radiotherapy and chemotherapy. Rather, a new area of focus is to reinvigorate the antitumor immunity of the host to resume its conquest against cancer cells through immunomodulatory therapies.

Radiotherapy is an essential treatment modality for many malignancies. Classically, radiation kills cancer cells through lethal DNA damage that leads to mitotic catastrophe or apoptosis. The four “R”s of classical radiobiology—reassortment, reoxygenation, repair, and repopulation—constituted the key elements for determining the success or failure of radiation treatment. However, recent advances in radiation biology have uncovered an intricate link between radiotherapy and the immune system. The coveted and previously mysterious abscopal effect, in which local tumor radiation triggers regression of a distant untreated lesion, is now attributed to radiation-induced systemic immune activation, a phenomenon that is underlain by the ability of radiation to elicit an immunogenic form of cell death. As such, Golden and Formenti perceptively postulated that the fifth “R” of radiobiology ought to be “tumor rejection” to highlight the immunological properties of radiotherapy [1]. However, radiotherapy by itself is rarely sufficient to overcome the immunosuppressing nature of tumor cells and the surrounding TME. In this booming era of immuno-oncology, a rational strategy is to utilize immunotherapy to bolster the immunogenic effects of radiotherapy, and vice versa.

In this review, we will discuss in detail the immunogenic mechanisms of radiation and its shortcomings as a stand-alone therapy. We will summarize the available preclinical and clinical evidence of synergy between radiation and various forms of immunotherapy. Furthermore, the importance of dose-fractionation and timing of radiotherapy will be highlighted. Finally, recent clinical trials evaluating the combination of radiation with immunotherapy will be elaborated, with an emphasis on optimizing dose and sequencing of radiotherapy in the combined setting. 

## 2. Pro-Immunogenic Effects of Radiation

The ability of radiation to interact with the immune system was known long before we understood the how and the why. The notion that tumor response to radiation is partly dependent on the availability of T-cells was first demonstrated nearly 30 years ago [2]. Since then, evidence of increased tumor-infiltrating lymphocytes (TILs) after irradiation has been well reported and firmly established the causal relationship between radiotherapy and antitumor T-cell response [3,4,5,6,7,8], which is preceded by dendritic cell activation via tumor antigen presentation [9,10]. 

Several mechanisms of radiation-induced immunogenic cell death have been brought to light over the past decade. Irradiated cells can release certain danger-associated molecular patterns (DAMPs), such as high mobility group box 1 (HMGB1), heat-shock proteins, uric acid, and adenosine triphosphate (ATP). HMGB1, heat-shock proteins and uric acid can activate toll-like receptor signaling, which leads to dendritic cell maturation and priming of cytotoxic T lymphocytes (CTLs) [11,12,13], while ATP acts through cell surface purinergic receptors to attract macrophages and activate dendritic cells [14,15]. Radiation can also induce the translocation of calreticulin from the endoplasmic reticulum to the cell membrane. Exposure of calreticulin on the tumor cell surface serves as a phagocytotic signal for dendritic cells and macrophages, thereby enhancing the immunogenicity of cell death [16].

Aside from modulating extracellular signaling at the time of cell death to increase immunogenicity, radiation also triggers several phenotypic changes in tumor cells to facilitate immune detection and tumor eradication. Radiation induces expression of several cell surface death receptors that enhances susceptibility of tumor cell death in the presence of immune cells expressing the corresponding ligands. Upregulation of FAS receptor by radiation can be engaged by its specific ligand FAS-L to trigger extrinsic caspase-dependent apoptosis [17,18]. Similarly, radiation-induced death receptor 5 (DR5) sensitizes cancer cells to apoptosis via binding of tumor necrosis factor-related apoptosis-inducing ligand (TRAIL) [19,20,21]. Furthermore, co-stimulatory molecule CD80 and stress-induced NKG2D ligand are promoted by radiation to facilitate tumor cell clearance by T-cells and NK-cells [22,23,24].

Immune-cell infiltration into the TME is a crucial component of antitumor immune response. One mechanism in which tumor vasculature can prevent immune cell extravasation into the TME is through downregulation of adhesion molecules on endothelial cells such as E-selectin, intercellular adhesion molecule (ICAM)-1/2, and vascular cell adhesion molecule (VCAM)-1 [25]. Notably, radiation has been shown to increase the expression of E-selectin and ICAM-1 in human endothelial cells [26,27], thus modifying the tumor vasculature to allow more robust immune-cell infiltration.

Immune-cell permeability across the tumor vascular endothelium is required but not sufficient for infiltration. Cytokine and chemokine milieu in the TME play a crucial role in homing of the immune cells into the tumor. Chemokine CXCL16, as well as CXCL9 and CXCL10 through interferon-γ signaling, is induced by radiation to promote recruitment of CD4+ and CD8+ T-cells [6,28,29]. Radiation also upregulates the release of pro-inflammatory cytokines, including interleukin-1 beta (IL-1β), tumor necrosis factor alpha (TNF-α), and type I and type II interferons (IFNs) to activate antitumor effects of infiltrated immune cells [30,31,32,33].

Efficacy of immune-mediated tumor killing largely hinges on the ability of T-cells to recognize tumor cells as rogue entities. In other words, effective antitumor immunity cannot be realized without robust immune response against tumor-specific antigens, even if successful tumor infiltration of lymphocytes and reversal of immunosuppressive checkpoint regulation were achieved. Indeed, a common mechanism of immune evasion deployed by cancer cells is downregulation of major histocompatibility complex class I (MHC-I) molecules for self-antigen presentation, thereby shrouding their aberrant genetic and phenotypic makeup [34]. Radiotherapy can counteract tumor immune evasion via several means. Irradiated tumor cells upregulate MHC-I to increase tumor detection by host immune system [35,36,37]. Furthermore, radiation damage may enhance presentation of tumor neoantigens to allow for robust antitumor immune activation through a combination of direct tumor cell killing, increased tumor mutational load from radiation-induced genetic instability, and radiation-dependent upregulation of specific tumor antigen expression. Together, this phenomenon is also known as in situ vaccination, which is further enhanced by radiation-induced stimulation of the innate immune system [38,39,40]. Collectively, radiation exhibits pro-immunogenic influences in the irradiated tumor in various aspects of host immune response against malignancy, as depicted in Figure 1.

## 3. Immunosuppressive Effects of Radiation

Despite having multiple pro-immunogenic properties, radiation can also augment several immunosuppressive effects. The most direct consequence of such is the depletion of antitumor lymphocytes within the irradiated tumor. Lymphocytes are exquisitely sensitive to the cytotoxic effects of ionizing radiation, with LD50 and LD90 (lethal dose of reducing surviving fractions of lymphocytes to 50% and 90%, respectively) of 2 Gy and 3 Gy, respectively [41]. Moreover, with conventional radiotherapy of delivering daily low dose radiation (1.8–2 Gy) over several weeks, mathematical modeling predicted significant radiation exposure to the circulating lymphocytes over the course of treatment, consistent with treatment-related lymphopenia that is commonly seen in irradiated patients [42]. In various cancers, treatment-induced lymphopenia is correlated with poor clinical prognosis, although whether or not this was due to compromised antitumor immunity is unclear [43]. In addition, preclinical studies have revealed differential effects of radiation among lymphocytes, with CTLs being more radiosensitive than regulatory T-cells (Treg). As such, radiotherapy may selectively deplete CD8 effector T-cells and proportionally increase Treg cells, which confer suppressive function within the TME to facilitate tumor escape from immunosurveillance [44,45].

Radiation also induces several immunosuppressive phenotypic changes in the TME through cytokine regulation. Tissue abundance of transforming growth factor-beta (TGF-β), an anti-inflammatory cytokine that suppresses intratumoral immune response, is increased with radiation [46,47]. Irradiated tumors also favor accumulation of immunosuppressive M2-polarized macrophages within the TME, with one study reporting tumor release of chemokine SDF-1α as one underlying mechanism [48,49]. Another study demonstrated that radiation increases expression of colony-stimulating factor 1 (CSF1) [50], a cytokine responsible for shifting macrophages towards M2 polarization and boosting the abundance of Treg and myeloid-derived suppressor cells (MDSCs), representing yet another signaling pathway to maintain the suppressive nature of TME.

Programmed death-ligand 1 (PD-L1), an immune checkpoint ligand that transmits an inhibitory signal to attenuate immune cell proliferation and activation, is upregulated on tumor and immune cells in the TME after irradiation. While one may view this as yet another suppressive attribute of radiation, evidence suggest that modulation of the PD-L1/PD-1 axis in response to radiotherapy may serve as a biomarker for antitumor immune activation. Dovedi et al. demonstrated that IFN-γ produced by activated antitumor CD8 T-cells was responsible for PD-L1 induction on tumor cells, representing an adaptive mechanism for cancer to thwart host reactive immunity [51]. By the same token, radiation can increase PD-1 expression on T-cells and weakens antitumor immunity [52]. However, this immunosuppressive sequela is preceded by successful mounting of T-cell responses against cancer cells, and PD-1 blockade offsets this countermeasure deployed by the tumor [51,52]. Consistent with these findings, PD-1 expression on TILs in HPV-positive head and neck cancer is a favorable prognostic marker and denotes antitumor immune activation after chemoradiotherapy [53]. 

## 4. Immune-Mediated Systemic Effects of Radiotherapy

Although radiotherapy is a local treatment, it has long been known to induce systemic effects. Documentation of distant tumor regression after local radiation exists as early as the beginning of the 20th century [54], a phenomenon also known as the abscopal effect. In 2004, Demaria et al. first provided evidence of an immune mechanism underlying the abscopal effect [5], now widely accepted as the culmination of the positive immune-mediated effects of local tumor radiation that primed the host’s immune system to eradicate distant non-irradiated disease of the same origin. Many clinical case reports of radiation-induced abscopal effect have been published over the past decades and are well-summarized in this recent review [55].

Nevertheless, the occurrence of the abscopal effect is exceedingly rare with radiotherapy alone. Given that radiation has both pro-immunogenic and immunosuppressive properties, its overall effect is dictated by the balance between the two opposing forces. While pro-immunogenic effects of radiation presumably dominate over its suppressive effects, a critical threshold of antitumor immunity is generally not realized in the absence of additional immunomodulation. Budhu et al. reported that a specified threshold of antigen-specific CD8 T-cells is required for efficient tumor killing in melanoma cell model, likely underscoring a prevalent challenge of insufficient immune activation with radiotherapy alone [56]. Unequivocally, in this era of increasing use of cancer immunotherapy, reports of abscopal effect have become more common [57]. As a prime example, a case report recounted a patient with metastatic melanoma with progressive disease on ipilimumab, received palliative radiation for a symptomatic paraspinal mass, and showed systemic disease regression after receiving an additional dose of ipilimumab two months after radiotherapy [58]. Accordingly, the crucial role of radiation in effective combinatory cancer therapy is increasingly being recognized.

In preclinical models, immune-mediated radiation effects generated long-lasting antitumor immunity. The development of radiation-induced immune memory is characterized by prolonged host survival and failure of tumor growth after subsequent rechallenge of the same tumor. Adoptive transfer of T-cells from successfully treated mice with radiotherapy into tumor-bearing mice led to tumor regression and extended survival [59,60]. At present, whether radiation alone can elicit persistent immune memory in the clinical setting is unclear. However, in line with the mechanism underlying the abscopal effect, long-lasting antitumor immune memory stimulated by radiotherapy is likely to become more apparent with rising utilization of immunomodulatory agents.

## 5. Synergy of Radiotherapy and Immunotherapy Combination

Because of the shortcomings of radiotherapy alone as a double-edged sword—having both immunogenic and immunosuppressive effects in the TME—addition of immunotherapy is a good strategy to overcome the inadequacy of radiation to mount a robust antitumor immune response. Preclinical and clinical evidence have demonstrated improved outcomes of radiotherapy in the presence of various types of immunotherapy that modulates different facets of tumor immunity.

### 5.1. Toll-Like Receptor Agonists

Toll-like receptor (TLR) signaling is crucial for activating dendritic cells to cross-prime effector T-cells. TLR agonists function to improve the ability of dendritic cells to present tumor antigens released from radiation cell killing. TLR9, the most extensively studied member of the TLR family, binds to unmethylated cytosine-phosphate-guanosine (CpG) oligodeoxynucleotide from bacterial DNA to induce cellular and humoral immunity. In murine model, TLR9 agonist has been shown to enhance therapeutic effects of radiation by increasing tumor-infiltration of natural killer dendritic cells, which led to fewer metastases and longer survival [61]. Similarly, targeting TLR9 with CpG oligodeoxynucleotide improved tumor response to radiation in preclinical models [62,63,64,65]. In vivo studies also demonstrated therapeutic synergy through combining TLR7/8 activation and irradiation, with combination treatment resulting in inhibition of local tumor growth and metastatic progression [66,67,68,69,70].

Clinically, combination of TLR9 agonist and radiation has shown some success in treatment of lymphomas. In a phase I/II study, 15 patients with stage III-IV relapsing low-grade B-cell lymphoma were treated with intratumoral injection with CpG DNA PF-3512676 and concurrent 4 Gy radiation to a single lesion, resulting in clinical response in 4 patients and stable disease regression in two [71]. In a subsequent phase I/II trial, injection with PF-3512676 was used in mycosis fungoides with 33% response rate, with clinical responders showing greater reduction in Tregs [72]. There are two trials evaluating the combination of TLR7 agonist imiquimod and radiation, one for breast cancer with skin metastases (NCT01421017) and another for diffuse intrinsic pontine glioma (NCT01400672), both with pending results. 

### 5.2. Cytokines

Cytokine signaling is the main mode of communication between immune cells to activate or suppress effector immune functions. Using pro-inflammatory cytokines to bolster effector cytotoxic T-cell functions can potentially overcome radiation-induced suppressive Treg accumulation. However, outcomes so far with combining cytokine therapy with radiation are modest at best. 

Interleukin-2 (IL-2) is a cytokine that regulates differentiation and proliferation of T-cells into effector and memory cells when stimulated by antigens. Preclinical studies on combining IL-2 and radiation are lacking. However, building upon the evidence that radiation can augment pro-inflammatory and immunogenic changes, a phase I study of stereotactic body radiation therapy (SBRT) in conjunction with high-dose IL-2 was performed in metastatic melanoma or renal cell carcinoma. Of the 12 patients treated, 8 patients had clinically significant response, and immune monitoring revealed greater proliferation of CD4+ T-cells with effector memory phenotype [73].

IL-12 is another pro-inflammatory cytokine that activates NK cells and cytotoxic CD8+ T-cells, as well as signaling differentiation of naïve CD4+ cells to T-helper 1 cells that can mediate antitumor immune response. Preclinical evaluation of combining radiation with IL-12 therapy is limited. However, one study showed that intratumoral expression of IL-12 led to increased IFN-γ levels and radiosensitizing effects [74]. There are no existing clinical trials testing the efficacy of this combination therapy.

Interferon-α (IFN-α) has broad immunological activities that modulate tumor immunity, including activation of dendritic cells and promotion of survival and expansion of natural killer (NK) cells and cytotoxic T-cells. IFN-α increases radiosensitivity of tumor cells in early in vitro studies [75,76]. A phase II trial showed improved survival in patients with resected pancreatic adenocarcinoma receiving adjuvant combination of chemoradiation and IFN-α compared to those receiving chemoradiation alone [77]. However, IFN-α therapy is highly toxic, leading to premature closure of phase II ACOSOG Trial Z05031 due to grade ≥3 toxicity of 95% [78]. Most recent phase III trial utilizing IFN-α in adjuvant chemoradiation for pancreatic cancer resulted in significant treatment toxicity without improvement in survival [79]. 

Tumor necrosis factor-α (TNF-α) is a potent inflammatory cytokine that has tumoricidal properties. However, earlier use of systemic TNF-α with concomitant radiotherapy has caused significant immune-related adverse effects and low patient tolerability [80]. TNFerade, a form of gene therapy in which human TNF-α gene controlled by a radiation-inducible promoter is delivered into cancer cells via replication-deficient adenoviral vector, has since been tested with radiotherapy in phase I/II trials with improved toxicity profile [81,82,83,84,85]. These promising results led to a phase III multicenter randomized trial for locally advanced pancreatic cancer patients treated with concurrent fluorouracil and radiation with or without intratumoral TNFerade. Despite being safe and well-tolerated, the addition of TNFerade did not improve overall or progression-free survival in this patient cohort [86]. 

A novel approach to cytokine therapy is to conjugate cytokines to antibodies or antibody fragments that specifically target tumor-associated antigens. The resulting class of fusion proteins, also known as immunocytokines, is capable of delivering cytokines directly to the tumor sites and avoiding systemic adverse effects that often limit the use of cytokine therapies [87]. Several recent studies have investigated the use of IL-2 immunocytokines with radiotherapy. L19-IL2, an immunocytokine with L19 antibody targeting the EDB-domain of fibronectin that is frequently overexpressed in solid tumors, has been shown to synergize with radiation against C51 murine colon carcinoma in a CD8-dependent manner [88]. Subsequent study by the same group demonstrated abscopal response in the non-irradiated lesions after L19-IL2 and radiation treatment to the index lesions, as well as long-lasting antitumor immunity in cured mice [89]. These encouraging results led to a phase I clinical study of combining L19-IL2 and SBRT in oligometastatic solid tumors (NCT02086721) with pending results. There are also recent reports of success with combining NHS-IL12 immunocytokine that targets necrotic cells with radiotherapy in preclinical models. Eckert et al. first showed that radiation-induced tumor necrosis can enhance intratumoral accumulation of necrosis-targeting NHS-IL12 immunocytokine, followed by a functional study demonstrating abscopal effect and improved survival in humanized mouse model bearing rhabdomyosarcoma xenografts [90,91]. While immunocytokines with TNF-α conjugates are also been tested in preclinical and clinical studies, combination of those agents with radiotherapy has yet to be studied.

### 5.3. Co-Stimulatory Molecules

After successful priming of tumor antigen-specific T-cells from antigen-presenting dendritic cells, co-stimulatory signaling are required to activate these T-cells to eradicate cancer cells harboring the corresponding antigens. There are two families of ligand/receptor proteins involved in T-cell co-stimulation: (1) B7/CD28 family that includes CD80/CD86 (B7-1/B7-2) ligand binding to CD28 receptor and CD275 (B7-H2) binding to CD278 (ICOS) receptor; and (2) TNF/TNF receptor family that includes ligands (CD40L, OX40L, CD70, and 4-1BBL) and its respective receptors (CD40, OX40, CD27, and 4-1BB). Intriguingly, cancer cells can evolve to hinder these essential stimuli via inhibitory signaling of T-cells (to be further discussed below under Section 5.4).

Recent in vivo studies have shown promising results of combining CD40 stimulation and radiation. Using a pancreatic ductal adenocarcinoma mouse model, Rech et al. revealed synergy between radiation and an agonist αCD40 antibody through distinct mechanisms. Ablative dose of radiation triggers early inflammatory stimulus through upregulation of MHC class I and CD86, while αCD40 causes a late response of altering intratumoral and systemic immunosuppressive myeloid cells, collectively yielding abscopal effect and long-term tumor immunity [92]. In another study using pancreatic cancer models, single fraction of SBRT with agonist αCD40 led to regression of non-irradiated tumor and durable immune memory [93].

OX40 is a potent co-stimulatory molecule on activated T-cells, and OX40 signaling can promote effector T-cell survival and inhibit Treg function, which can be achieved via OX40 ligand binding or stimulation via antibody agonists. Combination of single dose of 20 Gy and intratumoral delivery of activating OX40 antibody in murine lung cancer model resulted in CD8 T-cell dependent tumor killing and tumor immunity [94]. Gough et al. corroborated this finding in murine 3LL-tumor model using high doses of radiation and αOX40 antibody to achieve extended survival and decreased tumor recurrence compared to single treatments alone [95]. Recently, a combination of radiation and OX40 agonist has demonstrated efficacy in anti-PD-1-resistent murine lung tumors to inhibit local and systemic tumor growth [96]. 

4-1BB, also known as CD137, is the first member of the TNFR family identified as a potential target for cancer immunotherapy. Ligation of 4-1BB receptor on activated T-cells with 4-1BBL or antibody agonist prompts anti-apoptotic signaling to prevent activation-induced cell death and reverse T-cell tolerance [97]. When combined with 4-1BB agonist, antitumor effect of radiation has been shown to be enhanced. Treatment with 4-1BB antibody agonist with radiation in murine lung (M109) and breast (EMT6) carcinoma models significantly delayed tumor progression [98]. With the addition of cytotoxic T-lymphocyte-associated protein 4 (CTLA-4) blockade, the combination of 4-1BB activation and radiation further improved survival in GL261 murine glioma model, which is associated with greater infiltration of CD4+ and CD8+ lymphocytes [99]. Similarly, concomitant inhibition of PD-1 with radiation and 4-1BB agonist enhanced antitumor effect against human BRAF-mutant melanoma [100]. Interestingly, one study showed decreased off-target immune cell activation with 4-1BB aptamer compared to 4-1BB antibody when combined with radiation, while having similar therapeutic effect with both agents, suggesting potential differences in treatment toxicity of targeting the same receptor with varying forms of agonists [101]. 

To date, clinical experience with combining radiation with co-stimulatory molecules is limited. However, there are several ongoing clinical trials (discussed below, Section 7) that will shed light on the clinical utility of this therapeutic strategy. 

### 5.4. Immune Checkpoint Inhibition

Immune checkpoints are regulatory mechanisms that serve to prevent over-stimulation of activated T-cells, which can lead to autoimmunity. CTLA-4 and PD-1, also members of the B7/CD28 family, are expressed on activated T-cells to act as an “off” switch when bound by ligands CD80/CD86 and PD-L1/PD-L2, respectively. Cancer cells often over-express PD-L1/PD-L2 to exploit the intrinsic mechanism of T-cell inhibition. Antibody antagonists against CTLA-4 and PD-1/PD-L1 attenuate tumor-induced inhibitory signaling, thereby shifting towards T-cell stimulation and bolstering adaptive tumor immunity. Moreover, as previously mentioned, radiotherapy can upregulate PD-1/PD-L1 on tumor and immune cells in the TME, and as such, combining checkpoint inhibition with radiation may nullify this undesired immunosuppressive sequela. 

Demaria et al. first reported synergy of radiation and CTLA-4 blockade in a preclinical setting. Using the poorly immunogenic murine 4T1 mammary carcinoma model, this study showed that only combinatory treatment of radiation and CTLA-4 inhibition, but not either treatment alone, exhibited significant survival advantage over control. Furthermore, systemic antitumor immunity was provoked with combined treatment resulting in decreased lung metastases, which required the presence of CD8+ T-cells [102]. In a follow-up study by the same group, abscopal effect of combined radiation and CTLA-4 blockade was demonstrated using bilateral tumor models in which the unirradiated tumors displayed significant growth delay after irradiation of the primary tumors. This systemic effect is CD8-dependent and correlates with increased TILs and tumor-specific IFN-γ-producing T-cells in the unirradiated tumors [7].

The number of prospective clinical investigations assessing safety and efficacy of combined CTLA-4 inhibition and radiotherapy is on the rise. So far, only a few trials have reported results. In a phase I/II study, patients with metastatic castrate-resistant prostate cancer were treated with escalating doses of ipilimumab with or without radiotherapy. Maximum dose tested of 10 mg/kg ipilimumab with 8 Gy radiation to one to three bony metastases showed acceptable toxicity profile, with one-third of patients having stable disease or better [103]. Using the same ipilimumab and radiation dose, a phase III multicenter trial randomized 799 patients with metastatic prostate cancer to ipilimumab versus placebo after radiotherapy to osseous metastases. Although median overall survival of patients receiving ipilimumab only trended higher than those receiving placebo (11.2 vs. 10.0 months, *p* = 0.053), post-hoc subgroup analysis of patients with good prognostic features demonstrated significant survival benefit with ipilimumab (22.7 vs. 15.8 months, *p* = 0.0038) [104]. There are also several small prospective studies that reported abscopal responses and improved overall survival in metastatic melanoma patients treated with ipilimumab and radiation to brain and/or visceral metastases [105,106,107,108]. Of interest, a joint clinical and preclinical study in patients with metastatic melanoma implicated T-cell exhaustion from upregulation of tumor PD-L1 expression in the resistance towards treatment with radiation and CTLA-4 antibody. Treatment-induced increase in the PD-1/PD-L1 axis was reproduced in murine melanoma models, and the addition of PD-L1 blockade significantly improved response to radiation and CTLA-4 inhibition [109]. In addition, a recent clinical study shed light on plausible mechanisms underlying favorable responses to combination treatment with radiation and CTLA-4 blockade. A cohort of 39 patients with metastatic NSCLC were treated with radiotherapy to one metastasis with concurrent ipilimumab, with response rate of 18% and disease control in 31% of patients. Increase in IFN-β and T-cell receptor clonal dynamics predicted response to combination therapy, and further characterization of a single responder revealed expansion of two specific T-cell clones that target an immunogenic mutation on a radiation-induced gene, KPNA2. While this intriguing finding supports the hypothesis that radiotherapy can enhance neoantigen exposure to the host immune system, validation in an expanded cohort is warranted. Furthermore, the single-arm nature of this trial precludes the determination of the degree of contribution from either radiotherapy or ipilimumab in the observed immunological effects [110].

Several preclinical models have also revealed therapeutic synergy of radiation and PD-1/PD-L1 blockade. Treatment of radiation and anti-PD-1 antibody in mouse glioma model improved survival compared to either treatment modality alone. Combined treatment group exhibited increased tumor infiltration of cytotoxic T-cells and decreased Tregs, and glioma tumor cells rechallenged in treated mice failed to grow [111]. Deng et al. observed increase in PD-L1 expression in the TME after radiation, and the addition of PD-L1 inhibition augmented antitumor effect of radiation. Specifically, efficacy of combined treatment is dependent on CD8+ T-cells and correlated with reduction in the immunosuppressive MDSCs [8]. Similar findings were reported by Dovedi et al., which also revealed that upregulation of PD-L1 on tumor cells is induced by IFN-γ secretion from CD8+ T-cells [51]. Subsequent studies further demonstrated that PD-1 blockade enhanced antigen-specific and tumor-specific immunity triggered by radiation [112,113]. In a recent mechanistic study, the authors showed that both the preexisting resident T-cells and infiltrating lymphocytes after combination treatment contributed to tumor regression in in-field and out-of-field tumors [114]. While most studies examined the role of PD-1/PD-L1 axis inhibition in bolstering radiation efficacy, Wang et al. reported that radiotherapy can reverse tumor resistance towards anti-PD-1 therapy through induction of IFN-β and MHC-I expression on tumor cells [115]. Together, the above evidence underscored the importance of modulating the immune status within the TME to optimize efficacy of cancer therapy in the clinic.

Given that ample evidence lent support to the efficacy of combining PD-1/PD-L1 blockade with radiation, an abundant of clinical trials are now ongoing to investigate the utility of this combination in the clinical setting. Several phase I/II trials have established that concomitant PD-1/PD-L1 inhibition with radiotherapy is generally well-tolerated without dose-limiting toxicities [116,117,118,119]. Furthermore, with increasing adoption of ablative radiotherapy for treatment of multiple metastatic foci, Luke et al. demonstrated that multisite SBRT to up to four lesions followed by pembrolizumab within 7 days of SBRT completion was well tolerated [120]. Recently, the phase III PACIFIC trial that randomized locally-advanced unresectable non-small cell lung cancer (NSCLC) patients to either adjuvant durvalumab (anti-PD-L1 antibody) or placebo after chemoradiation demonstrated significant improvement in progression-free survival with durvalumab (median survival 16.8 vs. 5.6 months) [121]. Most recent update of the trial results also showed improved overall survival with patients receiving durvalumab compared to those receiving placebo (2-year overall survival 66.3% vs. 55.6%) [122]. It is interesting to note that the secondary analysis of KEYNOTE-001 trial, a study in which locally advanced or metastatic NSCLC patients were treated with anti-PD-1 pembrolizumab, showed improved survival in patients who received prior radiotherapy. Although this analysis was retrospective in nature and hypothesis-generating, it nevertheless shed light on the potential therapeutic synergy of radiotherapy and PD-1/PD-L1 blockade in the clinical setting [123].

### 5.5. Macrophage Polarization

Tumor-associated macrophages (TAMs) play important roles in tumorigenesis and contribute to maintaining an immunosuppressive TME in their default state. TAMs are typically pro-tumorigenic and phenotypically resemble M2 macrophages, and reducing tumor-infiltration of macrophages or modifying the polarity of immunosuppressive TAMs towards pro-inflammatory M1 phenotype have shown to impair tumor growth [124]. As previously discussed, radiation also promotes M2 polarization within the irradiated tissues. As such, by reversing the immunosuppressive phenotype of TAMs should augment the immunogenic effects of radiation. 

CSF1 is a key cytokine responsible for promoting M2 polarization through CSF1 receptor (CSF1R)-mediated signaling and CSF1R blockade in tumor models led to repolarizing of TAMs to the M1 phenotype [125,126]. Xu et al. provided evidence that CSF1R blockade improved efficacy of radiotherapy against prostate cancer in a murine model, with associated decrease in intratumoral MDSCs and TAMs populations [50]. Furthermore, recent study demonstrated that macrophages can express PD-1 and that PD-1 expression correlated with M2 polarization. Blockade of PD-1/PD-L1 axis enhanced phagocytosis of tumor cells by intratumoral PD-1+ macrophages in vivo and decreased tumor burden [127]. Given that checkpoint inhibition of PD-1/PD-L1 axis is already being widely used, this finding underscores a novel mechanism in which PD-1/PD-L1 blockade can bolster therapeutic effects of radiotherapy.

## 6. Effect of Radiation Dose and Timing on Immunogenicity

Dose fractionation and timing of radiotherapy are important attributes of treatment efficacy when combining radiation with various forms of immunotherapy. However, there is currently no clear evidence regarding the optimal dose and timing of radiation when utilized with immunotherapy in the clinical setting, highlighting the need for well-designed clinical trials to address this concern.

### 6.1. Dose per Fraction

Preclinical studies have shown that a wide range of doses per fraction can induce several immunogenic molecular changes in the TME. Major histocompatibility complex class I (MHC-I), which is crucial for antigen presentation from cancer cells to allow for tumor detection by host immune system, can be induced by single radiation doses of 8–25 Gy [29,35,36] or daily doses of 2 Gy/fraction to a total of 50 Gy [128]. Radiation of various doses can also upregulate anti-tumor cytokine expressions, including interferon-beta, interferon-gamma, interleukin-1-beta, chemokine CXCL16, and tumor necrosis factor-alpha [6,29,30,31,129].

Currently, more evidence points towards SBRT/hypofractionated doses as being more immunogenic, although most studies discussed here compared regimens with different biologically effective dose. In B16/OVA murine model, single fraction of 15 Gy to the tumor resulted in greater tumor control and increased activation and infiltration of antitumor T-cells compared to 3 Gy × 5 fractionated doses [3]. Comparing different fractions of delivering a total dose of 15 Gy, Schaue et al. demonstrated greatest tumor response with 7.5 Gy × 2, with associated increase in activated IFN-γ-producing T-cells and relatively low proportions of Tregs [130]. In the setting of combined therapy with CTLA-4 inhibition, Dewan et al. reported that 8 Gy × 3 regimen against TSA mouse breast carcinoma resulted in enhanced tumor response of both irradiated and non-irradiated tumors compared to the two other tested fractionations (20 Gy × 1 and 6 Gy × 5), with frequency of CD8+ T-cell activation proportional to treatment response [7]. Similarly, a single fraction of 12 Gy with concomitant PD-L1 blockade led to effective tumor control and antitumor modulation of immune cell milieu in the TME [8]. Recently, Vanpouille-Box et al. demonstrated systemic antitumor abscopal effect using combined treatment with 8 Gy × 3 and CTLA-4 blockade. Mechanistically, the authors showed that doses between 4 Gy to 12 Gy per fraction upregulates IFN-β production and secretion via the cyclic GMP-AMP synthase (cGAS) and its downstream stimulator of interferon genes (STING) pathway. Notably, doses of >12 Gy per fraction induced Trex1-mediated degradation of cytosolic DNA and abrogated the immunogenic secretion of IFN-β, illustrating that ablative doses of radiation, at least in certain cancer cells, may in fact negate the immunogenicity of tumor cell death [131]. While the evidence above collectively shows antitumor immunogenicity elicited with hypofractionated doses (and perhaps not ablative doses), it is important to note the wide spectrum of doses reported, suggesting that heterogeneity in the optimal dose-per-fraction likely exists among different tumor types. 

On the other hand, results for conventional fractionation with low doses per fraction are mixed. As discussed earlier, lymphocytes are very radiosensitive and conventional fractionation often leads to lymphopenia [41,42]. As a case in point, a study comparing SBRT to conventional fractionation radiotherapy in pancreatic cancer revealed that rates of severe lymphopenia were 13.8% versus 71.7%, respectively [132]. Some evidence has also demonstrated immunosuppressive properties of low-dose radiation. In an ex vivo model, macrophages receiving doses of 0.1–0.5 Gy exhibit anti-inflammatory status with reduction in IL-1β secretion and increase in TGF-β expression [133]. In another study, conventional fractionation of 2 Gy × 5 in several cell lines in vitro, compared to 10 Gy × 1, led to induction of TGF-β-associated and IFN-related genes that are conducive of an immunosuppressive TME [134]. On the contrary, other studies have successfully stimulated antitumor immunity with conventional fractionation. An in vivo study using low-dose radiation was effective for normalizing tumor vasculature, which would facilitate the migration of immune cells across the endothelium and into the tumor [135]. Furthermore, low-dose radiation induced M1 macrophage phenotype and subsequently increased T-cell recruitment into the irradiated tumor [135,136]. Two studies from the same group also demonstrated potent T-cell dependent antitumor response with fractionated 2 Gy × 5 with concurrent PD-L1 inhibition [51,114]. These conflicting data underscore the biological complexity of dose-fractionation, which is likely influenced by tumor histology and utilization of different immunotherapies. As an example in which tumor histology can affect radiation outcome, melanin in melanoma cells confer radioprotection by serving as free-radical scavengers, and as such, effective radiotherapy for melanoma typically requires higher dose-per-fraction [137,138]. Therefore, further studies are warranted to systematically identify the optimal doses in specific tumors in combination with specific immunomodulatory agents. 

### 6.2. Timing

There is limited evidence to guide the ideal timing of radiotherapy when used in conjunction with immunotherapy. Depending on the mechanism of actions of the immunomodulatory agents used, the optimal timing of radiotherapy relative to administering immunotherapy is likely to differ. 

In a preclinical model, Dovedi et al. determined that PD-L1 blockade is only effective when given concurrently with radiotherapy of 2 Gy × 5, but not sequentially two days after the five-fraction course [51]. This contrasts with the limited clinical evidence available. Subgroup analysis of the PACIFIC trial showed that progression-free survival is higher when patients received durvalumab within 14 days after chemoradiation compared to those who received treatment >14 days after (hazard ratio of 0.39 vs. 0.63) [121]. Secondary analysis of KEYNOTE-001 also reported the observation that patients receiving radiotherapy prior to pembrolizumab had improved median survival of 10.7 months vs. 5.3 months in those without previous radiotherapy, suggesting a possible temporal benefit when radiotherapy precedes PD-1/PD-L1 blockade [123]. Given that PD-L1 expression on tumor and immune cells are upregulated after radiation and serves as a mechanism of resistance by promoting T-cell exhaustion, as previously discussed, inhibition of the PD-1/PD-L1 axis shortly after radiotherapy appears to be reasonable. The contrasting evidence from the above preclinical model may potentially be attributed to the short subtherapeutic radiation regimen used (2 Gy × 5). However, it is possible that moving PD-1/PD-L1 blockade to the concurrent phase may further improve its therapeutic synergy with radiation and is now being investigated in a phase I trial CINJ 031507 in locally advanced NSCLC (NCT02621398).

Study by Young et al. also noted distinct effective radiotherapy timing with other immunotherapy agents. In mice bearing CT26 murine colorectal carcinoma, CTLA-4 blockade was most efficacious when given prior to, rather than after, focal radiation of 20 Gy. On the other hand, OX40 agonist antibody was most effective when administered one day after 20 Gy radiation, with decreased efficacy if given several days before or after radiation [139]. These findings are consistent with known mechanisms of CTLA-4 and OX40 therapies; CTLA-4 inhibition can deplete intratumoral Tregs prior to radiotherapy to mitigate the immunosuppressive TME and enhance immunogenicity of radiation, while OX40 co-stimulatory molecules are upregulated only for a brief period after antigen presentation induced by radiation [140,141]. Although challenging given the complex dynamics of the tumor immune response to combination treatment, further efforts are required to elucidate the optimal temporal relationship of various immunotherapies with radiotherapy in the clinical setting.

## 7. Ongoing Clinical Trials Assessing Combination of Radiotherapy with Immunotherapy

There are over a hundred clinical trials in the United States that were opened over the last two years designed to investigate different combinations of immunotherapies with radiation. Table 1 summarizes the relevant ongoing trials that were initiated after 1 May 2016; for a listing of earlier trials, see comprehensive reviews by Vacchelli et al. and Bloy et al. [142,143]. Many of the recent trials are phase I trials testing the safety and toxicity of combination therapies with various immunomodulatory agents and dose-fractionations of radiotherapy in different malignancies, which will not be included in Table 1 due to space limitation.

A significant proportion of recent trials utilize a combination of different immunotherapies with radiation, which partly stems from the realization that treatment response rates remain to be limited with radiotherapy combined with single immunotherapy agents (although still an improvement over radiotherapy alone). Inefficacy of combining with single agents could be attributed to the development of resistance against a given checkpoint blockade or the presence of multiple simultaneous immunosuppressive signals within the TME, which can be rescued if the appropriate additional agents were included in the regimens, as eloquently illustrated by Twyman-Saint et al. [109]. Another rationale for combining different immunomodulatory agents is improved therapeutic efficacy compared to single agents alone. In particular, a recent phase I trial NCT03431948 is testing the combination of 4-1BB agonist or CSF1R inhibitor with PD-1 blockade in conjunction with radiotherapy, which stems from preclinical evidence that the immune-activating potential of these agents is more robust in the combinatory setting [100,144]. Of note, many trials also included the use of chemotherapy, often given concurrently with radiotherapy, which may potentially alter the cumulative effects of combination therapy. Discussed separately in another review, certain chemotherapeutic agents can modulate the immune milieu of the TME or trigger immunogenic cell death, which should be considered when designing multi-modality trials [145].

However, key issues that are equally important to address lie within the intrinsic properties of radiation—timing of radiotherapy, dose-fractionation, and radiation modalities—as discussed in the previous section. Currently, only a handful of trials are designed to shed light on these subjects and are denoted accordingly in Table 1. Timing of immunotherapy relative to radiotherapy is being assessed in NSCLC patients receiving chemoradiotherapy and pembrolizumab (NCT02621398) and in a metastatic small cell lung cancer cohort treated with SBRT and nivolumab/ipilimumab (NCT03223155). Although both are phase I trials primarily evaluating the feasibility of shifting immunotherapy into the concurrent setting with radiotherapy, their secondary endpoints of comparing the efficacy and immunological changes of sequential vs. concurrent immunotherapy would provide valuable insight on the sequencing of different treatments in combination therapy. High versus low dose-per-fraction radiotherapy in combination with immunotherapy is being evaluated in two phase II trials (NCT02888743 and NCT03085719), which would attempt to address the potential differences in immunogenicity of fraction size clinically. Although whether radiation modality affects the immunogenic properties of radiotherapy is less clear and beyond the scope of this review, several phase I/II trials are testing the feasibility and outcomes of combining different radiation sources with immunotherapy (NCT02913417, NCT03486197, and NCT03325816). As we move forward with the strategy of enhancing radiotherapy through additional immunomodulation, we must address the uncertainty of how to best incorporate radiation in this multi-modality approach.

## 8. Conclusions

In summary, radiation can be considered as a form of immunological cancer therapy. While radiotherapy alone is unlikely to prevail against tumor evasion from the immune system, synergy of combining radiation with immunotherapy can better harness the immunogenic effects of radiotherapy. While evidence for the use of combination therapy in NSCLC and melanoma is more robust at the time of this review, many clinical trials are underway to ascertain the feasibility and efficacy of combining radiotherapy with immunotherapy in various malignancies. As we advance forward down this promising path of improving the ability to cure cancer with combination therapy, more efforts are necessitated to scrutinize the optimal dose and timing of radiotherapy in the combined setting.

## Figures and Tables

**Figure 1 cancers-10-00515-f001:**
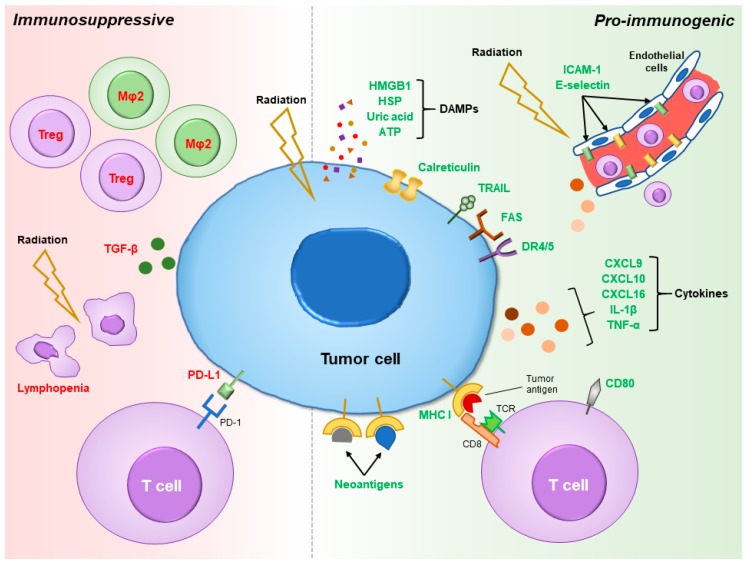
Pro-immunogenic and immunosuppressive properties of radiation. Radiation promotes tumor immunogenicity by release of danger-associated molecular patterns (DAMPs) to attract and activate immune cells, translocation of calreticulin to the cell surface to serve as a phagocytotic signal, upregulation of death receptors and ligands (TRAIL, FAS, and DR 4/5), release of various pro-inflammatory cytokines, increase in MHC I expression to facilitate antigen presentation, and neoantigen formation. Low-dose radiation to the tumor vasculature can also induce ICAM-1 and E-selectin expression on endothelial cells to promote extravasation of immune cells into the tumor microenvironment (TME). On the other hand, immunosuppressive effects of radiation include lymphopenia due to cytotoxic effects of radiation on lymphocytes, proportional increase in Tregs, M2 polarization of macrophages, secretion of anti-inflammatory cytokine TGF-β, and induction of PD-L1 expression on tumor cells. Abbreviations: HMGB1 = high mobility group box 1; HSP = heat shock protein; ATP = adenosine triphosphate; TRAIL = tumor necrosis factor-related apoptosis-inducing ligand; DR4/5 = death receptor 4/5; IL-1β = interleukin 1β; TNF-α = tumor necrosis factor α; MHC I = major histocompatibility complex class I; ICAM-1 = intercellular adhesion molecule 1; Mφ2 = M2 macrophage; TGF-β = transforming growth factor β; PD-L1 = programmed death ligand 1.

**Table 1 cancers-10-00515-t001:** Recently initiated clinical trials for combining radiotherapy and immunotherapy.

Immune Target	Indication	Drug	Phase	Status	Type of RT	Combined with Other Therapies	Clinical Trial #
**PD-1**	Breast	Pembrolizumab	II	R	EBRT	None	NCT03051672
	Carcinoma unknown primary	Pembrolizumab	II	R	HFRT	None	NCT03396471
	Esophago-gastric	Nivolumab	I/II	R	CFRT	Carboplatin/paclitaxel	NCT03278626
		Pembrolizumab	II	R	CFRT	Carboplatin/paclitaxel	NCT03064490
		Pembrolizumab	II	R	CFRT	Capecitabine	NCT03257163
		Pembrolizumab	II	R	EBRT	None	NCT02830594
	GBM	Pembrolizumab	II	NYR	CFRT	None	NCT03661723
		Pembrolizumab	II	R	CFRT	Temozolomide/HSPPC-96	NCT03018288
		REGN2810	I/II	R	EBRT	Temozolomide/gene therapy	NCT03491683
	HNSCC	Nivolumab	II	NYR	CFRT	None	NCT03715946
		Nivolumab	II	R	CFRT	None	NCT03521570
		Nivolumab	I/II	R	HFRT	None	NCT03247712
		Pembrolizumab	II	NYR	CFRT	None	NCT03383094
		Pembrolizumab	II	R	CFRT/HFRT	None	NCT03085719 ^d^
		Pembrolizumab	II	R	CFRT	None	NCT03057613
		Pembrolizumab	III	R	CFRT	Cisplatin	NCT03040999
		Pembrolizumab	II	R	CFRT	Cisplatin	NCT02777385
		Pembrolizumab	I/II	R	CFRT	Cisplatin	NCT02759575
	Hodgkin lymphoma	Pembrolizumab	II	R	CFRT	Multiple chemotherapy cocktails	NCT03407144
		Pembrolizumab	II	R	CFRT	None	NCT03179917
	Kidney	Nivolumab	II	R	SBRT	None	NCT02781506
	Melanoma	Pembrolizumab	I/II	R	CFRT	A-dmDT390-bisFv (UCHT1) immunotoxin	NCT02990416
	Merkel Cell carcinoma	Pembrolizumab	II	R	SBRT	None	NCT03304639
	Nasophary-ngeal carcinoma	Nivolumab	II	R	CFRT	Cisplatin	NCT03267498
	Lymphoma	Pembrolizumab	II	R	CFRT	None	NCT03210662
	NSCLC	Nivolumab	II	R	SBRT	None	NCT03110978
		Pembrolizumab	II	R	CFRT	Carboplatin/paclitaxel/cisplatin/pemetrexed	NCT03631784
		Pembrolizumab	II	R	CFRT	None	NCT03523702
		Pembrolizumab	II	R	SBRT	None	NCT03217071
		Pembrolizumab	I	R	CFRT	Carboplatin/paclitaxel	NCT02621398 ^t^
	Pancreatic	Nivolumab	II	R	SBRT	Cyclophosphamide/GVAX pancreas vaccine	NCT03161379
	Pediatric	REGN2810	I/II	R	CFRT/HFRT	None	NCT03690869
	Prostate	Nivolumab	I/II	R	CFRT + HDRB	ADT	NCT03543189
	Rectal	Pembrolizumab	II	R	CFRT	Capecitabine/fluorouracil/leucovorin/oxaliplatin	NCT02921256
	SCLC	Nivolumab	I/II	R	177Lu-DOTA0-Tyr3-Octreotate	None	NCT03325816 ^m^
		Pembrolizumab	II	R	CFRT	Cisplatin/carboplatin/etoposide	NCT02934503
	Sarcoma	Pembrolizumab	I/II	R	CFRT	None	NCT03338959
		Pembrolizumab	II	R	CFRT	None	NCT03092323
	Urothelial carcinoma	Nivolumab	II	R	CFRT	None	NCT03421652
		Pembrolizumab	II	NYR	Neutron EBRT	None	NCT03486197 ^m^
		Pembrolizumab	II	NYR	HFRT	None	NCT03419130
**PD-L1**	Cervical	Atezolizumab	II	NYR	SBRT	None	NCT03614949
	Esophago-gastric	Durvalumab	II	R	CFRT	Carboplatin/paclitaxel	NCT02962063
	GBM	Atezolizumab	I/II	R	CFRT	Temozolomide	NCT03174197
		Avelumab	II	R	HFRT	None	NCT02968940
	HNSCC	Avelumab	III	R	CBRT	Cisplatin	NCT02952586
	Lymphoma	Atezolizumab	II	R	CFRT	None	NCT03465891
	Mesothelioma	Avelumab	I/II	R	SBRT	None	NCT03399552
	Metastatic brain (breast)	Atezolizumab	II	R	SRS	None	NCT03483012
	NSCLC	Avelumab	I/II	R	SBRT	None	NCT03050554
		Durvalumab	II	NYR	SBRT	None	NCT03589547
		Durvalumab	I/II	R	SBRT	None	NCT03148327
		Durvalumab	II	R	SBRT	None	NCT02904954
	Ovarian	Avelumab	II	R	SBRT	None	NCT03312114
	Pancreatic	Durvalumab	I/II	R	SBRT	None	NCT03245541
	Urothelial carcinoma	Durvalumab	I/II	R	HFRT	BCG	NCT03317158
		Durvalumab	I/II	R	CFRT	None	NCT02891161
**Others**	Lymphoma	TLR9 agonist SD-101	I/II	R	CFRT	Ibrutinib	NCT02927964
**Combo**	Esophago-gastric	Nivolumab Relatlimab	I/II	NYR	SBRT	None	NCT03610711
	GBM	Avelumab Epacadostat	I/II	NYR	CFRT	Bevacizumab	NCT03532295
		Ipilimumab Nivolumab	II	R	HFRT	None	NCT03367715
	Hepato-biliary carcinoma	Durvalumab Tremelimumab	II	R	EBRT	None	NCT03482102
	HNSCC	Durvalumab Tremelimumab	I/II	R	SBRT	None	NCT03618134
		Durvalumab Tremelimumab	I/II	R	SBRT	None	NCT03522584
	Kidney	Ipilimumab Nivolumab	II	R	SBRT	None	NCT03065179
	Lymphoma	Anti-OX40 antibody BMS 986178 SD-101	I	R	CFRT	None	NCT03410901
		Epacadostat SD-101	I/II	R	EBRT	None	NCT03322384
	Melanoma	Ipilimumab Nivolumab	II	R	HFRT	None	NCT03646617
	Merkel Cell carcinoma	Ipilimumab Nivolumab	II	R	SBRT	None	NCT03071406
	NSCLC	Atezolizumab Nivolumab Pembrolizumab	II	R	HFRT/SBRT	None	NCT03176173
		Durvalumab Tremelimumab	II	R	CFRT	None	NCT03237377
		Intralesional IL-2 Nivolumab Pembrolizumab	I	R	HFRT	None	NCT03224871
		Ipilimumab Nivolumab	I/II	NYR	CFRT	Platinum-based chemotherapy	NCT03663166
		Ipilimumab Nivolumab	I/II	R	HFRT	None	NCT03168464
	Pancreatic	Cabiralizumab Nivolumab	II	R	SBRT	None	NCT03599362
	Prostate	Pembrolizumab SD-101	II	R	SBRT	Leuprolide/abiraterone/prednisone	NCT03007732
	Rectal	Epacadostat Pembrolizumab	I/II	NYR	CFRT	Capecitabine/oxaliplatin	NCT03516708
	Sarcoma	Durvalumab Tremelimumab	I/II	R	CFRT	None	NCT03116529
		Ipilimumab Nivolumab	II	R	CFRT	None	NCT03307616
	SCLC	Ipilimumab Nivolumab	I	R	SBRT	None	NCT03223155 ^t^
	Urothelial carcinoma	Durvalumab Tremelimumab	II	NYR	CFRT	None	NCT03601455
	Uveal melanoma	Ipilimumab Nivolumab	I/II	R	Yttrium 90	None	NCT02913417 ^m^
	Multiple sites	Atezolizumab Nivolumab	II	R	HFRT	None	NCT03115801
		Atezolizumab Nivolumab Pembrolizumab	II	R	HFRT	Nelfinavir	NCT03050060
		Atezolizumab Nivolumab Pembrolizumab	II	R	SBRT	None	NCT03313804
		Cabiralizumab Nivolumab Urelumab	I	R	SBRT	None	NCT03431948
		Durvalumab Tremelimumab	II	R	CFRT/HFRT	None	NCT02888743 ^d^
		Ipilimumab Nivolumab	II	R	EBRT	None	NCT03104439
		Pembrolizumab ADV/HSV-tk	II	R	SBRT	Valacyclovir	NCT03004183
		Pembrolizumab IL-2	I/II	NYR	HFRT	None	NCT03474497

Abbreviations: RT = radiotherapy; GBM = glioblastoma; HNSCC = head & neck squamous cell carcinoma; NSCLC = non-small cell lung cancer; SCLC = small cell lung cancer; TLR9 = toll-like receptor 9; IL-2 = interleukin 2; R = recruiting; NYR = not yet recruiting; EBRT = external beam radiotherapy (fractionation unspecified); HFRT = hypofractionated radiotherapy; CFRT = conventionally-fractionated radiotherapy; SBRT = stereotactic body radiotherapy; HDRB = high-dose rate brachytherapy; SRS = stereotactic radiosurgery; BCG = Bacillus Calmette–Guérin therapy. ^d^ Trials comparing different dose-fractionation, ^t^ Trials evaluating timing of immunotherapy, ^m^ Trials assessing unique radiation modalities combined with immunotherapy.

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
