# Peer review of "Radiotherapy as a New Player in Immuno-Oncology"

_cancers, 2018, doi:10.3390/cancers10120515_

Reviewer 1 Report

This is a fairly exustive review. A few additions could make it more timely

Adding the current published works demonstrating safety (line 327, Luke et al, JCO 2018, Formenti et al, Nat Med 2018) and discussion of the biology in those works and how that relates to the animal models would be helpful.

The abstract is awkward - "However, the immunological efficacy of radiotherapy, as a double-edged sword, is limited by the concomitant inducti" - please reword.

 It may be helpful to also include the ongoing trials of the agents discussed. For example, it is stated for 4-1bb "as in ongoing trials discussed below". However the SBRT/Nivo/Urelemab trial form BMS is not listed (C4 MOSART #NCT03431948 that is nearing completion)

The cursory talk of macrophages or polarization needs expansion  - with CSF1r being the only target that likely not to be dropped from BMS developmetn portfolio. This is an odd ommission (see C4 MOSART). This has broad implications for IBC, Pancrease and others and is the the only agent outside of IPI this reviewer can think of that seems to have ever shown enhacement in any setting over single angent checkpoint alone (SITC 2018).

TNFerade seems a bit off topic here. The failure of that trial is complex and may have zero to do with TNF (did they even prove 1ug was made??)?

Author Response

Reviewer #1

1)      This is a fairly exhaustive review. A few additions could make it more timely.

Thank you very much for this positive comment. Your suggestions are very helpful and certainly have improved this manuscript. The specifics of the changes are described below.

2)      Adding the current published works demonstrating safety (line 327, Luke et al, JCO 2018, Formenti et al, Nat Med 2018) and discussion of the biology in those works and how that relates to the animal models would be helpful.

Thank you for this suggestion. A discussion on Luke et al., JCO 2018 is added at lines 396-398 to provide evidence of safety for administering multisite SBRT with pembrolizumab. A discussion on Formenti et al., Nat Med 2018 is added at lines 362-373 to describe the biological basis of synergy between radiation and ipilimumab.

3)      The abstract is awkward - "However, the immunological efficacy of radiotherapy, as a double-edged sword, is limited by the concomitant induction" - please reword.

Thank you for this criticism. This sentence has been reworded at lines 8-10.

4)      It may be helpful to also include the ongoing trials of the agents discussed. For example, it is stated for 4-1bb "as in ongoing trials discussed below". However the SBRT/Nivo/Urelemab trial form BMS is not listed (C4 MOSART #NCT03431948 that is nearing completion)

Thank you for pointing out this important omission. Trial NCT03431948 is now added to Table 1. Furthermore, discussion on the relevance of this trial is added at lines 542-546.

5)      The cursory talk of macrophages or polarization needs expansion - with CSF1r being the only target that likely not to be dropped from BMS development portfolio. This is an odd omission (see C4 MOSART). This has broad implications for IBC, Pancreas and others and is the only agent outside of IPI this reviewer can think of that seems to have ever shown enhancement in any setting over single agent checkpoint alone (SITC 2018).

Thank you for this critical suggestion. As stated above, NCT03431948 is now added to Table 1, and the importance of testing PD-1/4-1BB and PD-1/CSF1R combinations with radiation is discussed at lines 542-546. In addition, a section on macrophage polarization is added at lines 409-426, which briefly describes the significance of tumor-associated macrophages (TAMs) in tumorigenesis and the roles of targeted therapies against CSF1R and PD-1/PD-L1 in reversing the immunosuppressive M1 polarization of TAMs and improving efficacy of radiotherapy.

6)      TNFerade seems a bit off topic here. The failure of that trial is complex and may have zero to do with TNF (did they even prove 1ug was made??)?

Thank you for this comment. The inclusion of TNF-α and TNFerade (lines 257-266) is to provide available evidence on combining cytokine-related immunomodulation with radiotherapy. Per the authors’ understanding, the severity of adverse effects from administering systemic TNF-α therapy has led to the development of its inducible counterpart, TNFerade. However, results from the most recent phase III trial with TNFerade were negative. Reviewer #1 is correct to point out the complexity of the failure of this trial. The inclusion of this trial was only meant to provide completeness of the discussion on TNF-α. However, if Reviewer #1 feels strongly that this content should be removed from the review manuscript, the authors are happy to comply.

Reviewer 2 Report

Review

The authors summarize the immune effects of cancer irradiation, abscopal effects and clinical approaches of combinatorial treatment strategies. The concepts described in the review article have been extensively discussed over the last years in the radiation oncology community. As the first reports and concepts have been published appr. 10 years ago (e.g. „Systemic effects of local radiotherapy“, Formenti and Demaria, Lancet Oncol 2009), radiotherapy is not really a „new player in immuno-oncology“ any more. Pubmed lists at least 12 review articles with similar contents published in 2018, not counting Reviews focussing on certain tumor entities, certain immunotherapy approaches etc. So the novelty of the summarized concepts seems somehow limited.

Specific comments:

-       Page 2, line 58: „immunogenic cell death have BEEN brought to light over“

-       Page 2, line 59: „Irradiated CELLS can release certain…“

-       Page 2, line 63: ATP actS throught cell surface puirnergic receptors“

-       Page 2, line 80ff: Hot and cold tumors most probably differ in their microenvironment, cytokine profile etc more than in the vasculature. There might be an indirect connection through hypoxia which is highly immunosuppressive.

-       Page 2, line 92: „CD4+ and CD8+ T-cells“

-       Page 3, line 96ff: Immunotherapy is a topic separate from effects of irradiation (heading oft he chapter)

-       Page 3, line 100: „effective antitumor immunity is unlikely in the absence of immunogenic presenting tumor antigen“ Please rephrase, not clear.

-       Page 3, line 102ff: „To counteract this, radiation damage may unmask previously undetected tumor neoantigens to allow for robust antitumor immune activation, a phenomenon also known as in situ vaccination, which is further enhanced by the above radiation-induced pro-inflammatory adjuvants.“ Not clear. Are neoantigens „unmasked“? More presentation due to MHC-I upregulation? More neoantigens due to genomic instability? „Adjuvants“ is normally used for artificial compounds used with antigens for vaccination. Maybe rephrase to „stimulation oft he innate immune system“?

-       Page 3, line 116: The role of lymphopenia in radiation therapy is not clarified. There is a correlation between lymphopenia and outcome in retrospective patient analyses. However, causality is not established. For certain immunotherapies (such as vaccination strategies or T-cell transfer), lymphopenia inducing drugs are used to enhance efficacy of immunotherapy.

-       Page 4, line 131: MAthematical modelling does not demonstrate biologic phenomena but more predict.

-       Page 4, line 134: The causality between lymphopenia, anti-tumor immune response and outcome is not clear (see above)

-       Page 4, line 140: „TransformING growth factor beta“, why add active?

-       Page 4, line 142: M2 polarization cannot be accumulated. Either „accumulation of M2 polarized macrophages“ or „favouring M2 polarization“

-       Page 5, line 175: Cells are not measured in „concentration“

-       Page 5, line 195: „addition of immunotherapy is a good strategy to overcome“

-       Page 5, line 204: „(NK) AND dendritic cells“?

-       Page 5, line 198ff: There are reports on Imiquimod and radiotherapy, which should be added.

-       Page 5, line 216ff: There are strategies to target cytokines tot he TME (e.g. immunocytokines employing IL2, IL12, TNF, nanoparticles etc) which should be added.

-       Page 6, line 267: „activating OX40 antibody“

-       Page 7, line 276: „antitumor effect of radiation has been shown tob e enhanced“

-       Page 7, line 281: Not clear what exactly the combination treatment was fort he BRAF mutated melanoma.

-       Page 7, line 291: „Immune checkpoints are regulatory mechanisms that serve to…“

-       Page 7, line 304: „combined treatment resulting in…“

-       Page 8, line 327: „significantly improved response to radiation and CTLA-4 inhibition“

-       Page 9, line 374: „chemokine CXCL16“

-       Page 9, line 377: 15 Gy and 3Gy x 5 are not comparable in their radiobiologic effect either (e.g. comparison of EQD2 or BED for both regimens).

-       Page 10, line 429: PD-L1 after irradiation showing an effect does not prove that simultaneous immune checkpoint inhibition might be even better in the clinical setting as well as suggested by preclinical data.

-       Table: Lists 177Lu-DOTA as treatment for SCLC. To my knowledge it is used for differentiated neuroendocrine tumors. Please check.

-       Table legend: BCG treatment ist he Bacillus Calmette-Guerin bacteria and not a vaccine

-       Page 14, line 499: „radiotherapy alone is unable to prevail against tumor evasion from the immune system. Not clear, part oft he radiation-effect might be immune dependent as shown by radiobiology experiments in immunocompetent versus immunosuppressed mice.

Author Response

Reviewer #2

1)      The authors summarize the immune effects of cancer irradiation, abscopal effects and clinical approaches of combinatorial treatment strategies. The concepts described in the review article have been extensively discussed over the last years in the radiation oncology community. As the first reports and concepts have been published appr. 10 years ago (e.g. „Systemic effects of local radiotherapy“, Formenti and Demaria, Lancet Oncol 2009), radiotherapy is not really a „new player in immuno-oncology“ any more. Pubmed lists at least 12 review articles with similar contents published in 2018, not counting Reviews focusing on certain tumor entities, certain immunotherapy approaches etc. So the novelty of the summarized concepts seems somehow limited.

Thank you for this criticism. While there are several reviews on topics such as immunogenic properties of radiotherapy and combination therapy with radiotherapy and immunotherapy, this invited review is meant to provide a comprehensive update on the landscape of radiotherapy in immuno-oncology for the special issue of “New Developments in Radiotherapy”. The merging field of radiotherapy and immunotherapy is moving at an ever-changing pace with continuous emergence of exciting new discoveries, and this review seeks to capture this. Discussion on the practice-changing PACIFIC trial, novel biological insights on combining radiotherapy with CTLA-4 or PD-1 blockade, and the therapeutic implication of PD-1 expression on tumor-associated macrophages are examples of important developments that have only emerged over the past year. With any comprehensive reviews, discussion of new developments is preceded by older evidence to contextualize their significance. Furthermore, this review devotes a significant focus on the issue of timing and dose-fractionation of the radiotherapy in the setting of combination therapy, which has not yet been well-addressed in the design of future clinical trials for combination therapy. Finally, a unique aspect of this review is the discussion of select new trials of combination therapy with radiation and different immunomodulatory agents that may address important issues as the oncology community move forward with this exciting new direction.

2)      Page 2, line 58: „immunogenic cell death have BEEN brought to light over“

Thank you for this correction. The above is corrected at line 59.

3)      Page 2, line 59: „Irradiated CELLS can release certain…“

Thank you for this correction. The above is corrected at line 60.

4)      Page 2, line 63: ATP actS throught cell surface puirnergic receptors“

Thank you for this correction. The above is corrected at line 64.

5)      Page 2, line 80ff: Hot and cold tumors most probably differ in their microenvironment, cytokine profile etc more than in the vasculature. There might be an indirect connection through hypoxia which is highly immunosuppressive.

Thank you for this comment. To avoid any misunderstanding, we have removed lines 82-84 regarding hot and cold tumors.

6)      Page 2, line 92: „CD4+ and CD8+ T-cells“

Thank you for this correction. The above is corrected at line 93.

7)      Page 3, line 96ff: Immunotherapy is a topic separate from effects of irradiation (heading of the chapter)

Thank you for this comment. The sentence of question has been modified at lines 97-98.

8)      Page 3, line 100: „effective antitumor immunity is unlikely in the absence of immunogenic presenting tumor antigen“. Please rephrase, not clear.

Thank you for this comment. The above sentence has been modified at lines 99-102 for clarification.

9)      Page 3, line 102ff: „To counteract this, radiation damage may unmask previously undetected tumor neoantigens to allow for robust antitumor immune activation, a phenomenon also known as in situ vaccination, which is further enhanced by the above radiation-induced pro-inflammatory adjuvants.“ Not clear. Are neoantigens „unmasked“? More presentation due to MHC-I upregulation? More neoantigens due to genomic instability? „Adjuvants“ is normally used for artificial compounds used with antigens for vaccination. Maybe rephrase to „stimulation of the innate immune system“?

Thank you for this comment. To make clear, a part of this paragraph is modified at lines 107-114 to address the ability of radiation to enhance presentation of tumor antigens/neoantigens and rephrasing of “stimulation of the innate immune system”.

10)   Page 4, line 131: MAthematical modelling does not demonstrate biologic phenomena but more predict.

Thank you for this correction. The above is corrected at line 140.

11)   Page 4, line 134: The causality between lymphopenia, anti-tumor immune response and outcome is not clear (see above)

Thank you for astutely pointing out the inadequacies of the data on radiation treatment and lymphopenia. Sentence at lines 142-144 is modified to reflect the uncertainty of causality between radiation and lymphpenia.

12)   Page 4, line 140: „TransformING growth factor beta“, why add active?

Thank you for this correction. The above is corrected at line 150.

13)   Page 4, line 142: M2 polarization cannot be accumulated. Either „accumulation of M2 polarized macrophages“ or „favouring M2 polarization“

Thank you for this correction. The above is corrected at lines 152-153.

14)   Page 5, line 175: Cells are not measured in „concentration“

Thank you for this correction. The above is changed to “specified threshold” at line 186.

15)   Page 5, line 195: „addition of immunotherapy is a good strategy to overcome“

Thank you for this correction. The above is corrected at line 206.

16)   Page 5, line 204: „(NK) AND dendritic cells“?

Thank you for pointing out the confusion. The authors intended to write natural killer dendritic cells as a single entity, and therefore, removed “(NK)” to avoid confusion at line 215.

17)   Page 5, line 198ff: There are reports on Imiquimod and radiotherapy, which should be added.

Thank you for this comment. Preclinical data on imiquimod and other TLR7/8 agonists are briefly mentioned at lines 218-220, and a sentence describing the two available clinical trials on combining radiotherapy with imiquimod is added at lines 226-229.

18)   Page 5, line 216ff: There are strategies to target cytokines to the TME (e.g. immunocytokines employing IL2, IL12, TNF, nanoparticles etc) which should be added.

Thank you for this comment. Strategies to target cytokines to the TME using immunocytokines are now discussed in lines 267-280. For completeness, a brief discussion on IL-12 is also added in lines 242-247.

19)   Page 6, line 267: „activating OX40 antibody“

Thank you for this comment. The above has been modified at line 301.

20)   Page 7, line 276: „antitumor effect of radiation has been shown to be enhanced“

Thank you for this correction. The above is corrected at line 311.

21)   Page 7, line 281: Not clear what exactly the combination treatment was for the BRAF mutated melanoma.

Thank you for pointing out the confusion. Clarification has been added at line 316.

22)   Page 7, line 291: „Immune checkpoints are regulatory mechanisms that serve to…“

Thank you for this correction. The above is corrected at line 325.

23)   Page 7, line 304: „combined treatment resulting in…“

Thank you for this correction. The above is corrected at line 339.

24)   Page 8, line 327: „significantly improved response to radiation and CTLA-4 inhibition“

Thank you for this correction. The above is corrected at line 362.

25)   Page 9, line 374: „chemokine CXCL16“

Thank you for this correction. The above is corrected at line 440.

26)   Page 9, line 377: 15 Gy and 3Gy x 5 are not comparable in their radiobiologic effect either (e.g. comparison of EQD2 or BED for both regimens).

Thank you for astutely pointing this out. Unfortunately, many of the available studies comparing high vs. low doses per fraction do not account for the large differences in BED. Clarification of this point is added at lines 442-443.

27)   Page 10, line 429: PD-L1 after irradiation showing an effect does not prove that simultaneous immune checkpoint inhibition might be even better in the clinical setting as well as suggested by preclinical data.

Thank you for this important point. Secondary analysis of KEYNOTE-001 is hypothesis-generating by nature, and available studies at this time cannot conclusive determine the causality and therapeutic benefit of different sequencing of radiation and PD-1/PD-L1 blockade. Here, we are simply generating hypotheses based on limiting available data and bridging current evidence with future developments underway. We do not know whether simultaneous immune checkpoint inhibition might be better, and hence testing the various sequencing of radiotherapy with checkpoint inhibitors would shed some light on this issue (such as the NCT02621398 discussed).

28)   Table: Lists 177Lu-DOTA as treatment for SCLC. To my knowledge it is used for differentiated neuroendocrine tumors. Please check.

Thank you for raising this question. After double-checking on ClinicalTrials.gov, NCT03325816 is listed for recruitment of patients with extensive-stage small cell lung cancer.

29)   Table legend: BCG treatment is he Bacillus Calmette-Guerin bacteria and not a vaccine

Thank you for this correction. The above is corrected at lines 532-533.

30)   Page 14, line 499: „radiotherapy alone is unable to prevail against tumor evasion from the immune system. Not clear, part of the radiation-effect might be immune dependent as shown by radiobiology experiments in immunocompetent versus immunosuppressed mice.

Thank you for pointing out this confusion. We meant to state that radiotherapy alone is unlikely to elicit a strong enough antitumor immune response due to the various intrinsic immunosuppressive effects of radiation, and thus warranting the utilization of combination therapy with immunomodulatory agents. Modification is made at line 572 to hopefully clarify this.

Reviewer 3 Report

This is a timely and for the most part well-written review on the important topic, which is appropriate for this journal. Some considerations are recommended.

Main

When mentioning melanoma, mention melanin and melanogenesis as important features of melanoma and melanocytes (Physiol Rev 84, 1155-1228, 2004). Mention that melanin and melanogenesis can regulate behavior of melanoma cells (Arch Biochem Biophys 563:79-93, 2014) and attenuate the response to radiotherapy (Exp  Dermatol 24: 258-259, 2015; Oncotarget 20:17844-1785, 2016 Feb 3. doi: 10.18632/oncotarget.7528; Int J Mol Sci 19(4), 1048; https://doi.org/10.3390/ijms19041048.).

Optional

You can mention that tumors at stage 3-4 disease, can not only regulate its own environment but also body homeostasis through immune-endocrine messengers produced by tumors (Mayo Clin Proc 89, 429-433, 2014). This concept can relate to any progressing malignancy, not only melanoma. Gamma and other type of radiation can affect these interactions at local and systemic levels in manner similar to ultraviolet (see for discussion: (Endocrinology 159(5), 1992-2007, 2018), since they are example of physical stressors.

Author Response

Reviewer #3

This is a timely and for the most part well-written review on the important topic, which is appropriate for this journal. Some considerations are recommended.

Thank you very much for the positive feedback.

Main

When mentioning melanoma, mention melanin and melanogenesis as important features of melanoma and melanocytes (Physiol Rev 84, 1155-1228, 2004). Mention that melanin and melanogenesis can regulate behavior of melanoma cells (Arch Biochem Biophys 563:79-93, 2014) and attenuate the response to radiotherapy (Exp  Dermatol 24: 258-259, 2015; Oncotarget 20:17844-1785, 2016 Feb 3. doi: 10.18632/oncotarget.7528; Int J Mol Sci 19(4), 1048; https://doi.org/10.3390/ijms19041048.).

Thank you for suggesting this very relevant point. Indeed, tumor histology is one of the major factors in governing the efficacy of radiotherapy, and the mechanism of melanin production in protecting melanoma cells from radiation is a prime example of this concept. Discussion of this is added in lines 481-484 to elude to the complexity of determining optimal dose-fractionation in the setting of combination therapy, as there is no one-size-fits-all regimen for different tumors.

Optional

You can mention that tumors at stage 3-4 disease, can not only regulate its own environment but also body homeostasis through immune-endocrine messengers produced by tumors (Mayo Clin Proc 89, 429-433, 2014). This concept can relate to any progressing malignancy, not only melanoma. Gamma and other type of radiation can affect these interactions at local and systemic levels in manner similar to ultraviolet (see for discussion: (Endocrinology 159(5), 1992-2007, 2018), since they are example of physical stressors.

Thank you for bringing up this fascinating concept. The example of melanoma being capable of utilizing neuroendocrine and melanogenesis to affect changes in systemic homeostasis and potentially hindering the therapeutic effects of radiotherapy, immunotherapy, and chemotherapy is very powerful and worthy of future exploration in the field of radiotherapy and immune-oncology. However, because we have already exceeded the 6000-word limit, and this complex concept is worthy of a review on its own, we hesitate to begin this discussion without being able to close it in a terse and informative manner. However, if Reviewer #3 feels strongly in incorporating this concept in specific section(s) of this manuscript, we would be happy to accommodate.

Round  2

Reviewer 2 Report

The authors have responded to the comments made on the manuscript. The new sections require some more modifications.

-       Page 3, line 107: “To counteract this,…” sounds like tumor cells “want” to counteract the immunosuppression. Please rephrase.

-       Page 3, line 113: “which is further enhanced by radiation-induced stimulation of…”

-       Page 7, lines 278ff: There are two papers reporting on the combination of the immunocytokine NHS-IL12 and irradiation (Eckert et al., Cancer Immunol Immunother 2016; Eckert et al., Oncoimmunol 2017) showing increased binding of the immunocytokine in irradiated tumors and immune responses and abscopal effects in a humanized mouse model.

-       Page 9, lines 411ff: The authors first state that TAMs are immunosuppressive, but later state that they can be repolarized to another phenotype that impairs tumor growth. Please rephrase to avoid inconsistencies.

Author Response

Reviewer #2 (2nd round)

The authors have responded to the comments made on the manuscript. The new sections require some more modifications.

Thank you for taking the time and effort to further refine this manuscript. Details of further revisions are described below and are highlighted green in the manuscript.

1)      Page 3, line 107: “To counteract this,…” sounds like tumor cells “want” to counteract the immunosuppression. Please rephrase.

Thank you for pointing this out. It should now be made clear that it is in fact radiotherapy that is counteracting the ability of tumor cells to evade immune surveillance (modification at line 107).

2)      Page 3, line 113: “which is further enhanced by radiation-induced stimulation of…”

Thank you for this suggestion. The sentence at line 113 is modified to reflect the above change.

3)      Page 7, lines 278ff: There are two papers reporting on the combination of the immunocytokine NHS-IL12 and irradiation (Eckert et al., Cancer Immunol Immunother 2016; Eckert et al., Oncoimmunol 2017) showing increased binding of the immunocytokine in irradiated tumors and immune responses and abscopal effects in a humanized mouse model.

Thank you for this suggestion. The two studies mentioned above regarding NHS-IL12 and radiation combination are now discussed in lines 277-282.

4)      Page 9, lines 411ff: The authors first state that TAMs are immunosuppressive, but later state that they can be repolarized to another phenotype that impairs tumor growth. Please rephrase to avoid inconsistencies.

Thank you for pointing out this potential source of confusion. We added clarification that TAMs in their default state (line 412) resemble M2 macrophages and contribute to the immunosuppressive nature of TME. However, these immunosuppressive TAMs can have their polarity modified through certain immunomodulatory agents (ie. CSF1R inhibitors) to exhibit antitumor effects (line 414).